# Resolution Potential of Necrotic Cell Death Pathways

**DOI:** 10.3390/ijms24010016

**Published:** 2022-12-20

**Authors:** Anett Mázló, Yidan Tang, Viktória Jenei, Jessica Brauman, Heba Yousef, Attila Bácsi, Gábor Koncz

**Affiliations:** Department of Immunology, Faculty of Medicine, University of Debrecen, Egyetem Square 1, 4032 Debrecen, Hungary

**Keywords:** necrosis, inflamamtion, resolution, cell death, SPM

## Abstract

During tissue damage caused by infection or sterile inflammation, not only damage-associated molecular patterns (DAMPs), but also resolution-associated molecular patterns (RAMPs) can be activated. These dying cell-associated factors stimulate immune cells localized in the tissue environment and induce the production of inflammatory mediators or specialized proresolving mediators (SPMs). Within the current prospect of science, apoptotic cell death is considered the main initiator of resolution. However, more RAMPs are likely to be released during necrotic cell death than during apoptosis, similar to what has been observed for DAMPs. The inflammatory potential of many regulated forms of necrotic cell death modalities, such as pyroptosis, necroptosis, ferroptosis, netosis, and parthanatos, have been widely studied in necroinflammation, but their possible role in resolution is less considered. In this review, we aim to summarize the relationship between necrotic cell death and resolution, as well as present the current available data regarding the involvement of certain forms of regulated necrotic cell death in necroresolution.

## 1. Introduction

Infections, following the recognition of pathogens by pathogen-associated molecular patterns (PAMPs), trigger inflammatory processes that eliminate foreign invaders. In the absence of pathogens, sterile inflammation can be induced by the release of damage-associated molecular patterns (DAMPs). The most common form of sterile inflammation initiated when DAMPs are released from dying cells is the so-called necroinflammation process [1]. Inflammation is followed by tissue regeneration or resolution to repair tissue damage as a consequence of pathogens or immune reactions [2].

In comparison to PAMPs, tissue damage does not necessarily indicate infection. According to this, dying cell-released factors not only amplify the inflammatory response to eliminate the pathogen, but they may also be a signal for tissue regeneration. The presence of resolution-associated molecular patterns (RAMPs) help counterbalance the inflammatory effects of DAMPs when they are released from damaged cells; their predominant cellular effects are the inhibition of inflammatory processes, immune regulation, and resolution of inflammation [3,4]. Dying cell-derived DAMPs and RAMPs are surveilled by cells with inflammatory or immunosuppressive potential in the dynamic tissue environment, resulting in the modulation of the immune response. This is achieved by the production of inflammatory cytokines and mediators or by resolving factors, primarily specialized pro-resolving mediators (SPMs) [2].

It is now clear that the pathway of cell death determines the outcome of the immune response. Apoptosis is the normal, primary cell death process in the body. Since it rarely induces unwanted tissue destruction, it is usually not accompanied by DAMP production; thus, it can be considered a non-inflammatory type of cell death. As a result, it does not activate tissue regeneration in most cases. On the other hand, after necrotic cell death processes, damaged tissues need to be restored. In recent years, many new necrotic and simultaneously regulated cell death pathways have been discovered. This includes cell death modalities such as pyroptosis, necroptosis, ferroptosis, parthanatos, mitochondrial permeability transition, entosis, netosis, oxeiptosis, and cuproptosis [5,6]. Recently, we have begun to recognize that these forms of necrotic cell death pathways differ not only in their triggering factors, morphology, and signaling pathways, but also in the mechanisms involving DAMP secretion, which subsequently results in an array of immunological outcomes (Figure 1). How the newly discovered cell death pathways contribute to resolution has not yet been deeply analyzed.

Sterile inflammation and consequent resolution should be considered a multistep process; sequentially: cell death, DAMP/RAMP release, tissue-specific sensing of these mediators, and the consequent release of pro-inflammatory/anti-inflammatory factors [7]. Current anti-inflammatory therapies are unable to prevent the causes of this multi-step process or inhibit the necrotic cell death pathways and dysregulated secretion of DAMPs/RAMPs. Understanding the necroinflammatory and necroresolving processes will thus enable more accessibility in the targeted intervention of these successive steps in the future. While readers can find comprehensive reviews on how different cell death modalities induce DAMP production [8], how cell types and their subpopulations modulate inflammation [9], and how the regenerative capacity of different tissue types modulates immunogenicity [10], the connection between cell death and RAMP secretion or resolution is hardly studied. This review will focus on how the factors released from dying cells regulate the balance between inflammation and resolution.

## 2. Proresolving Roles of Intracellular Proteins

The activation of innate immunity through different pattern recognition, such as pathogen-, damage-, microbe-, or function-associated molecular patterns (PAMP, DAMP, MAMP, FAMP) [11] has been a widely accepted principle. In the last decade, numerous theories emerged regarding the pattern-related inhibition of innate immunity. The presence of normal own structures, such as self- (SAMP) [12,13] or homeostatic-associated molecular patterns (HAMP) [14,15], ensures a continuous inhibitory signal for the natural immune system. Suppressive (SAMP) [7] or inhibitory (iDAMP) [16] molecules released during cell damage can counteract DAMP-mediated inflammation. Resolution-associated molecular patterns, RAMPs, were also able to counterbalance DAMP- or PAMP-induced tissue destruction [4,17]. We propose using the RAMP terminology, as it was one of the first to be introduced [17], and others, such as the SAMP abbreviation, have been used for many other purpose, such as for self-suppressive-symbiont-stress-associated molecular patterns [18,19]. The function and nomenclature of these molecular groups have not been clearly categorized yet. Blocking the activation of innate immunity or the function of DAMPs, inducing resolution, or executing resolution may imply partially overlapping, but even significantly different functions. Here, we aim to explore the role of RAMPs in cell death-induced resolution. We will not review the effect of self-associated molecular patterns continuously present on the surface of living cells. Readers are referred to comprehensive reviews of this topic elsewhere: [12,13].

Several ubiquitously expressed intracellular proteins are from necrotic cells to orchestrate resolution. The extracellular appearance of **HSP10** mediates immunosuppressive functions. It antagonizes LPS-induced inflammatory cytokine production whilst simultaneously increasing the production of anti-inflammatory IL-10. Its functional activity is beneficial in the immunotolerant phase of pregnancy, the resolution of tissues in periodontitis, and the alleviation of many autoimmune diseases [17,20]. As HSP proteins lack a classical secretion signal sequence, it can be assumed that HSP10 secretion is similar to the classical DAMPs released during necrosis, such as HSP70 and HSP90 [20,21], However, it is not yet clear how the functions of pro vs. anti-inflammatory HSPs are separated because several receptors have been identified that can recognize different HSP proteins overlapping, such as CD91, CD40, CD36, CD14, toll-like receptors (TLRs), and scavenger receptor-A [22].

**HSP27** has been published as a pro-inflammatory molecule in human coronary endothelial cells, activating TLR-2 and TLR-4 receptors, but its immunosuppressive and anti-inflammatory effects have also been demonstrated. Recombinant HSP27 stimulation resulted in increased IL-10 production in monocytes, inducing differentiation of anergic T cells, but it did not increase IL-1 and TNF production, and monocytes treated in this way induced differentiation of anergic T cells. The immunosuppressive role of HSP27 is supported by the fact that HSP27 treatment has been shown as a possible therapeutic option for atherosclerosis [23]. Additionally, extracellular HSP27 in the tumor microenvironment contributes to the immune evasion of tumors by inducing the differentiation of tolerogenic macrophage subpopulation and enhancing angiogenesis through VEGF production [17]. The ubiquitous expression of HSP27 is upregulated in various stress processes [24], such as during necrosis, resulting in the release of HSP27. HSP27 has been shown to be stored in exosomes or lysosomes, as well, and can be released by exosomes [23].

The anti-inflammatory role of **αB-crystallin** (HSPB5), which also belongs to the small heat shock protein family, is known in various auto-inflammatory diseases. It vitro stimulation of macrophages with αB-crystallin-derived peptide reduced the production of inflammatory cytokines (IL-1β, IL-6, IL-12, TNF-α), and treatment with recombinant αB-crystallin suppressed astrocytes and microglia-mediated inflammatory response [25]. In vivo αB-crystallin protected retinal ganglion cells against ischemia/reperfusion [26] and attenuated the inflammatory response in injured spinal cords [27] and experimental autoimmune encephalomyelitis [28]. The immunosuppressive function of αB-crystallin is also supported by its tumor-promoting role, although whether this is due to its extracellular or intracellular functions has not been investigated [29,30]. As with HSP27, αB-crystallin is upregulated during stress conditions [24], and may also be secreted by cells via exosome production [31].

Immunoglobulin heavy-chain-binding protein (**BiP**) (also known as glucose-regulated protein 78 (Grp78), or HSPA5) is an endoplasmic reticulum-related chaperone. Its expression is upregulated in times of cellular stress, and its BiP level is elevated during apoptosis [32], necroptosis [33], pyroptosis [34], and ferroptosis [35]. The extracellular appearance of BiP mediates anti-inflammatory effect, inducing IL-10 production and simultaneously inhibiting TNFα and IL-6 secretion by PBMC cells. It also upregulates soluble receptor antagonists of IL-1 and TNF-receptor [36]. It promotes the development of tolerogenic DCs and regulatory T cells, as well as reduces the expression of costimulatory receptors on antigen-presenting cells. BiP has been shown to induce long-term therapeutic protection in disease models of inflammatory arthritis [37], and its secreted form plays a role in angiogenesis and tumor progression [32]. It has a cell protective role in apoptosis, pyroptosis, and ferroptosis, but, surprisingly, inhibition of BiP significantly suppressed cigarette smoke-induced necroptosis [38].

**AnnexinA1** (ANXA1) is a phospholipid-binding protein expressed in innate immune and epithelial cells, whose expression is regulated by glucocorticoids. It is classified as a DAMP because it activates the formyl peptide receptors (FPR) responsible for the detection of bacterial and mitochondrial formylated proteins. ANXA1 is also known to stimulate T cell activation and promote Th1 cell differentiation [39]. However, ANXA1, especially when released from apoptotic neutrophils, regulates anti-inflammatory responses and induces tissue resolution. It inhibits PLA2, thereby blocking synthesis of prostaglandins and leukotrienes and production of IL-6, IL-1β, and TNF-α. It does, however, result in the production of IL-10 and TGFβ. In addition, it restricts leukocyte recruitment, facilitates engulfment of cellular debris, pushes macrophage differentiation towards the M2 phenotype, limits mast cell degranulation, and activates wound repair and muscle regeneration. In pathological processes, it has a protective role in chronic inflammatory disorders, and its expression is upregulated in different types of cancers [40,41]. ANXA1 mimetic peptide also accelerates wound healing [42]. ANXA1 is located on the inner surface of the plasma membrane and lacks a signal peptide. The molecule could be released as a component of microparticles and extracellular vesicles, especially from gelatinase granules of the neutrophil [42]. Its expression is also increased in necrotic processes; hence, it can also be passively secreted by destroyed cells [43].

Some publications have also referred to cardiolipin, a phosphatidylglycerol lipid molecule, as a RAMP [44]. Accordingly, stimulation with unsaturated, but not saturated, cardiolipin blocks LPS-induced NF-κB activation and TNF-α and IP-10 secretion in human and murine macrophages, as well as LPS-induced TNF-α and IL-1β release in human blood mononuclear cells [44]. However, cardiolipin as mitochondrial-DAMP has been shown to block resolution and cause persistent inflammation [45].

## 3. A Brief Summary of the Resolution

Tissue-localized immune cells are characterized by their abundance of inflammatory and tolerogenic cell types. These cell types refer to the DAMPs and/or RAMPs released from dying cells, which induce the production of either pro-inflammatory or pro-resolution mediators. As the regulatory role of DAMPs in the production of inflammatory mediators and cytokines is well known, the next section will briefly review the main functions of SPMs, lipoxins, resolvins, maresins, and protectins and describe how their production is regulated.

SPMs are biosynthesized in response to chemicals produced by acute inflammatory responses. Their role as ”resolution agonists” actively coordinates the resolution of inflammation. Without effective resolution, homeostasis will not occur [17]. They are generated from polyunsaturated fatty acids (arachidonic acid/AA, eicosapentaenoic acid/EPA, docosapentaenoic acid/DHA) by lipoxygenase enzymes via transcellular mechanisms. Synthesis of maresin and protectins requires 12- lipoxygenase (12-LOX) and 15-LOX-1, respectively, while multiple lipoxygenases are also involved in lipoxin and resolvin synthesis (summarized in Table 1).

Early inflammatory events instigate SPM generation and consequent resolution. Lipid mediator class switching changes AA metabolism from producing proinflammatory mediators to producing SPMs, functioning as an endogenous resolution program. For example, the proinflammatory mediators’ prostaglandin E2 (PGE2) and prostaglandin D2 can elevate 15-LOX gene expression levels, leading to the conversion of leukotriene A4 to lipoxin. At the site of inflammation, neutrophil-platelet interactions facilitate the transfer of lipid mediator precursors from one cell type to another, resulting in the transcellular biosynthesis of SPMs [46]. The recruited neutrophils die following extravasation, which is followed by efferocytosis of apoptotic polymorphonuclear neutrophils (PMNs) by macrophages, stimulating subsequent SPM production. Microparticles from PMNs also stimulate macrophages to produce SPMs (e.g., resolvin D5, protectin D1) and PGE2 [47].
ijms-24-00016-t001_Table 1Table 1The role of the different specialized pro-resolving mediators in the resolution.
LipoxinResolvin EResolvin DProtectin/NeuroptotectinMaresinPrecursorAAEPADHADHADHAReceptorsALX/FPR2 [48]ChemR23 [49]ALX/FPR2GPR32 [50]GPR37 [51]RORaLGR6 [52]Synthesis
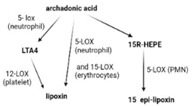

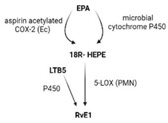

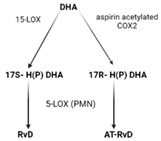

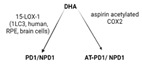

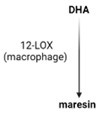
Effects1. ↑neutrophil apoptosis2. ↓ cytokine production from neutrophil and T cell3. ↓ macrophage apoptosis4. ↑ macrophage efferocytosis5. ↓ B cell Ab production1. ↓ transendothelial and transepithelial migration of PMN2. ↑ clearance of PMNs from apical mucosal surface [53]3. ↓ growth of Gram- bacteria [54]4. ↑ Macrophage efferocytosis [55]5. ↓ DC IL12 production and migration towards T cell zone in spleen [49]6. ↓ inflammatory pain [56]1. ↓ neutrophil infiltration [57,58]2. ↑ macrophage efferocytosis3. ↓ T cell cytokine production and differentiation to Th1 and Th17 [50]4. ↑ generation of Treg5. ↓ inflammatory pain [56]6. ↑ bacteria clearance [58,59]PD:1. ↓ PMN infiltration2. ↓ T cell cytokine production3. ↑ T cell apoptosis [60]NPD: ↓ RPE apoptosis from oxidative stress [61]1. ↓ PMN infiltration2. ↑ macrophage efferocytosis3. ↑ M2 polarization4. ↓ pain5. ↑ tissue regeneration [62]

Interestingly, certain traditional anti-inflammatory therapies, such as Nonsteroidal Anti-Inflammatory Drugs (NSAIDs), are considered “resolution disruptors”. They inhibit the biosynthesis of eicosanoids involved in the initiation of inflammation, thereby impacting the generation of SPMs essential for resolution. In contrast, aspirin can trigger the production of various types of SPMs via the acetylation of COX2 in endothelial cells. Irreversible acetylation of the COX-2 catalytic site blocks prostaglandin synthesis. However, the acetylated enzyme is not completely inactivated and can convert AA to biosynthetic intermediates of 15-epi-LXA4, also termed aspirin-triggered Lipoxin A4 [46].

Briefly, SPMs mainly function in enhancing macrophage efferocytosis, as well as inhibiting neutrophil and eosinophil infiltration and migration of DCs to the lymph nodes. These molecules block histamine and proinflammatory cytokine production, specifically helper T-cell cytokine production (IFNy, IL4, IL-17, IL-23), inhibiting their ability to stimulate the immune response. In parallel, SPMs induce the differentiation of immune tolerogenic cells, such as anti-inflammatory M2 macrophages, stimulate anti-inflammatory cytokine and amphiregulin production, and support tissue regeneration and wound healing (reviewed in [2,63]).

## 4. Turning Points in Resolution

The composition of tissue, along with its basic regenerative capacity, significantly impacts the transition from inflammation to resolution. Cells, such as macrophages, dendritic cells, fibroblasts, stem cells, neutrophil granulocytes and mast cells, either localized within the inflamed tissue or recruited, can differentiate into tolerogenic subtypes at the site of inflammation. These sensor cells produce pro-inflammatory or pro-resolving factors influenced by dynamic tissue conditions. The evolution of inflammation towards the resolution phase is initiated by mediators originating from dying cells, especially during sterile inflammation. However, the released DAMPs and RAMPs are directly regulated in the tissue environment by post-translational modifications and co-working signaling pathways that can also influence their effect (Table 2). In this section, we will review how subsequent changes in factors released from dying cells affect resolution.

High mobility group box 1 protein (**HMGB1)** is one of the most extensively studied DAMPs. This protein activates both classical PRRs and DAMP-specific receptors, resulting in a proinflammatory effect similar to LPS. Its secretion has been detected during essentially every cell death process. However, signals mediated by HMGB1 can have anti-inflammatory or pro-resolving functions, which are achieved by the simultaneous co-signaling of inhibitory receptors or via post-translational modifications. Inhibitory molecules, such as CD24, CD52, or C1q, can form signaling complexes with HMGB1, altering its effects. CD24 functions by forming a tri-molecular complex with Siglec-10 and either DAMPs, such as HMGB1, or heat shock proteins, such as HSP70 or HSP90; the inhibitory effect of Siglec-10 reduces the complex’s proinflammatory activity [64]. Deficiencies of CD24 or Siglec-10 result in increased proinflammatory cytokine production upon HMGB1 stimulation, but their absence does not modify the cytokine production of DCs in response to LPS. Soluble CD52 also facilitates the binding of HMGB1 to the inhibitory receptor Siglec-10, converting the pro-inflammatory functions of HMGB1 into an anti-inflammatory mechanism [65]. A tetramolecular complex of HMGB1 with C1q, RAGE, and LAIR-1 directs monocyte differentiation into an anti-inflammatory phenotype [66]. Mechanistically, HMGB1-plus-C1q stimulates signaling that regulates the phosphorylation and localization of 5-LOX; consequently, nuclear transport of 5-LOX is inhibited, and 5-LOX accumulation in the cytosol leads to the increased production of SPMs. In addition, the HMGB1-C1q complex also induces elevated MerTK expression, leading to increased SPM production. This complex inhibits the expression of proinflammatory cytokines, but increases the production of the immunosuppressive cytokine, IL-10, and programmed death ligand 1 (PD-L1), which is a suppressive co-stimulatory molecule involved in the T cell response [67].

Post-translational modifications also affect HMGB1, of which oxidation appears to be the most important in regulating immunological outcomes. HMGB1 contains three highly conserved cysteines (C23, C45, and C106) that are susceptible to reduction or oxidation. Three known forms of HMGB1 include fully reduced HMGB1, disulfide HMGB1 containing partially oxidized cysteines 23 and 45, and sulfonyl HMGB1, in which all the cysteines are terminally oxidized. Oxidized variations of HMGB1 bind to different receptors, determining the bioactivity of extracellular HMGB1. Fully reduced HMGB1 induces chemotaxis, the disulfide HMGB1 variation can stimulate proinflammatory cytokine secretion, while sulfonyl HMGB1 remains in its inactivated state [68]. In general, the extracellular medium is highly oxidizing compared to the cytoplasm, and the regulation of the pH has a strong effect on inflammatory processes through the modification of HMGB1. Overall, HMGB1 secretion does not necessarily provoke inflammation; the type of cell death, the receptors present on the cell surface during efferocytosis, and the tissue environment together determine the immunomodulatory effect of HMGB1.

**ATP** released from dying cells activates the innate and adaptive immune system. Its high expression during necrotic cell death processes has been published [69]. Secretion of ATP may result in the exacerbation of necroinflammation because its extracellular form induces pyroptosis [70]. However, articles published on ATP have shown that it also contributes to liver regeneration, angiogenesis, and wound repair with the simultaneous release of anti-inflammatory proteins, such as ANXA1 [71,72]. Importantly, extracellular ATP is turned into adenosine by the cooperation of CD39 (ATP→AMP) and CD73 (AMP→adenosine) ecto-enzymes [73]. As a result, the ATP-driven proinflammatory environment switches to an anti-inflammatory environment upon adenosine production. Adenosine induces the differentiation of immune suppressive cells, such as M2 macrophages, Treg, or myeloid-derived suppressor cells (MDSCs), suppresses the innate immune response, and has a direct effect on tissue regeneration and wound healing. Through negative feedback, the concentration of extracellular adenosine drastically increases in inflammatory conditions [73]. Tissue-specific capacity for adenosine production [74] and the expression level of CD39 and CD73 has a deterministic influence on the ATP-mediated immunoregulation [73].

**IL-33**, considered a conventional DAMP, is known primarily for its pro-inflammatory role. Its contribution to tissue resolution has been published, as well [75,76]. IL-33 is a constitutively expressed full-length protein in the nuclei of endothelial cells, mucosal epithelial cells, and fibroblasts; however, in macrophages and dendritic cells, this cytokine is weakly or not expressed at all [77]. Similar to IL-1a, full-length IL-33 is a bioactive molecule that does not strictly require maturation. Because IL-33 is translated without a signal sequence, it is not secreted by the classical ER-Golgi pathway [78], but rather is rapidly released from necrotic cells as a classical alarmin.

The expression of the IL-33 receptor, ST2, is restricted to mast cells, type 2 innate lymphoid cells (ILC2), Th2, and regulatory T cells [79]. Through these cells, IL-33 activates the key regulator of the Th2 arm of the immune response, among others, leading to the production of IL-4 and IL-13. These interleukins strongly support the differentiation of M2 macrophages [79]. IL-33-activated ILC2 cells produce amphiregulin, which directly contributes to resolution [80]. In addition, IL-33 is an activator of mast cells, strengthening type II inflammation, and may also have a role in pro-inflammatory cytokine production in macrophages [79], as well as facilitating neutrophil extracellular trap formation [81]. In vivo studies confirm the multifaceted role of this cytokine; IL-33 is an important regulator of tissue repair in tissues, such as the skin, muscle, liver, kidney, and intestine. It also regulates central nervous system injuries and supports wound healing [82,83], along with limiting the effects of obesity-associated inflammation [76], and it is important for endotoxin tolerance [84]. In contrast, it can worsen pathology in colitis [83] and type two inflammatory diseases, such as allergy, asthma, and atopic dermatitis [85,86,87]. Thus, IL-33 can mediate a pro- or anti-inflammatory response in a tissue-dependent manner, the direction of which is mostly determined by the cell types that detect this alarmin.

Posttranslational modifications may also regulate IL-33-mediated responses. IL-33 is proteolytically cleaved by caspases during apoptosis [88], although its inactivation by caspase-1 is controversial [88,89]. Serine proteases released by granulocytes and mast cells cleave IL-33, thereby increasing its activity considerably.

Direct interaction of IL-33 with other regulators of resolution has also been published. IL-33 increases the expression of CD73 and CD39, which are involved in ATP-adenosine transition [90]. MerTK and 15-LOX-1 expression was also regulated in an IL-33-dependent manner [91,92], while 5-LOX and 12-LOX contributed to IL-33-induced synthesis of IL-13 in bone marrow-derived mast cells [93].

The biological response of **ANXA1** is mediated by the FPR1 and FPR2/ALX receptors, which primarily recognize classic PAMPs, such as bacterial and mitochondrial formylated peptides, all of which will mediate a pro-inflammatory response. Furthermore, ANXA1 has various roles as a pro-resolving mediator [94]. Different signaling outcomes depend on the cell types activated by ANXA1. ANXA1 is a well-known activator of immunogenic cell death, acting on DCs and promoting DC-mediated cross-presentation that triggers a cytotoxic T-cell response [95]. In contrast, ANXA1 pro-resolving functions are mostly connected to macrophages and neutrophil granulocytes. Two different receptors may also explain the contradictory behavior of ANXA1. FPR1 is mostly mentioned as an activator of immunogenic cell death or human polymorphonuclear neutrophils [96]. FPR2, on the other hand, has more pro-resolution activity, supported by the fact that the pro-resolving mediators lipoxin A4 and resolvin D1 are also FPR2 ligands. However, the functional separation of the two receptors does not provide a completely sufficient explanation. The classical DAMP, serum amyloid A, is an inflammatory ligand of FPR2 [97]. Here, we must consider that other receptors, recognizing the same ligand, can provide co-signals that may modify the function of FPR2. Accordingly, serum amyloid A, but not ANXA1, may activate other receptors, as well, including TLR2 and TLR4, class B scavenger receptor CD36, and the ATP receptor P2X7 [98].

The different ligand preferences of FPR1 and FPR2 and the structural variations of these receptors may also be responsible for either promoting or resolving inflammation [99]. A single FPR2 receptor is capable of binding to pro-resolving or pro-inflammatory agonists with different receptor domains [100]. Multiple conformational changes of the receptor upon binding different ligands may explain the switch between proinflammatory and anti-inflammatory cell responses [101].

Enzymatic cleavage of ANXA1 also regulates its inflammatory effect. The full-length protein or the N-terminal 26-amino acid proved to be anti-inflammatory, and it is the Calpain 1-dependent cleavage that converts ANXA1 to its proinflammatory [102]. Serine proteases also inactivate its anti-inflammatory properties [103].

**PGE2** is a prostanoid produced by a sequence of actions mediated by COX enzymes and microsomal PGE2 synthase 1 [104,105]. It can be produced by fibroblasts, stromal, epithelial, and immune cells.

The level of PGE2 peaks can be also detected during inflammation and resolution, indicating its role in both phases. Prostaglandin exerts its effect on inflammation by recruiting PMNs and manipulating their effects. PGE2 mediates resolution by changing neutrophils into their pro-resolution phenotype, which stops the production of chemoattractants, such as leukotrienes B4. The peak of PGE2 production occurs before the peak of lipoxin A4 production since PGE2 signaling increases 15-LOX expression, converting arachidonic acid into lipoxins [106,107,108]. At the site of inflammation, apoptotic PMNs have higher PGE2 biosynthesis, suggesting these cells can initiate lipid class switching and resolution [47].

In rheumatoid arthritis, PGE2 has a pro-resolution impact by upregulating the nuclear translocation of p50 and downregulating p65. It also increases IkBa expression, therefore, preventing activation of NF-kB [109]. As NFκB activation leads to COX2 synthesis and ultimately prostaglandin synthesis, it seems plausible that this inhibition by PGE2 serves an important role in regulating the negative feedback of its own synthesis [110], which can make PGE2 production cyclical.

PGE2 effects are mediated by E-type prostanoid receptors (EP1-EP4). PGE2 receptors coupled to different G-proteins induce different immunological reactions. Through EP2/EP4 on DCs, PGE2 stimulates IL-23 production, which induces the formation of Th17, while through EP3 receptors, PGE2 activates mast cells. Utilizing these mechanisms, PGE2 plays a role in the pathogenesis of immune disorders [111]. In contrast, EP4 activation has an anti-inflammatory effect, e.g., inhibition of TNFa production from glial cells after LPS stimulation [112].

The effects of prostaglandin are both cell- and context-specific. They depend on the balance between its COX2-regulated synthesis, 15-hydroxyprostaglandin dehydrogenase-driven degradation, and the expression pattern of PGE2 receptors. The expression of specific cytokines and chemokines, as well as their cognate receptors present in immune cells, stromal cells, and epithelial cells, determines the effect of PGE2 in the transition from inflammation to resolution.

**5-LOX** is an essential enzyme in both leukotriene and SPM production. Its perinuclear localization has pro-inflammatory effects through the induction of leukotriene production, whilst cytoplasmic localization of 5-LOX induces resolution by reducing leukotriene production and promoting SPM production. The cytoplasmic accumulation of 5-LOX can also affect the simultaneous COX2 activity, where co-signaling of these two mediators leads to SPM production, further enhancing the anti-inflammatory role of 5-LOX [113]. The location and function of 5-LOX are determined by the phosphorylation of 5-LOX at its phosphorylation sites, namely on ser271, ser663, and ser523. Phosphorylation at ser271 by MAP kinase-activated protein kinase2 (MK2) [114] and at ser663 by ERK1/2 [115] expose nuclear import sequences inducing nuclear translocation of 5-LOX, leading to inflammatory effects. In contrast, MerTK and resolvin D1 inhibit 5-LOX nuclear transport, which downregulates ser271 phosphorylation, by controlling the activity of Calcium/calmodulin-dependent protein kinase II, p38, mitogen-activated protein kinase, and MK2 [116]. Phosphorylation of 5-LOX at ser523 by protein kinase A inhibits 5-LOX nuclear translocation [117].
ijms-24-00016-t002_Table 2Table 2Mediators with pro-inflammatory and pro-resolution activity.
Inflammatory RoleAnti-Inflammatory RoleHMGB11. Reduced HMGB1 induces chemotaxis2. Disulfide bond HMGB1 stimulates proinflammatory cytokine secretion [68]1. HMGB1-Siglec10-CD24 or CD52 tri-molecular complex [64]2. HMGB1-C1q-RAGE-LAIR1 tetramolecular complex -> 5LOX phosphorylation -> SPM production; ↑ MerTK expression -> ↑ IL10, PDL1, ↓ proinflammatory cytokines [65,66,67]ATP1. ↑ Pro-inflammatory cytokines2. Cross-presentation of DC -> ↑initiation of CTL response [69]3. Exacerbate inflammation by inducing pyroptosis [70]1. ↑ Macrophage release of anti-inflammatory proteins2. CD39 and CD73 degrade ATP to adenosine -> differentiation of M2, Treg, MDSCs; ↑ tissue regeneration and ↑ wound healing [73]IL331. Activate mast cell2. ↑ Neutrophil extracellular traps formation [79]1. Activate ILC2 -> produce amphiregulin [80]2. ↑ IL5,13 production -> ↑M2 differentiation [79]3. Regulate MerTK expression [91]4. ↑ 15-LOX1 expression [92]5. ↑ CD73, CD39 expression [90] ANXA11. Cross-presentation of DC -> ↑ initiation of CTL response [95]2. Act on FPR1 [96]3. Calpain1-dependent cleavage converts ANXA1 to proinflammatory [102]1. Act on FPR2/ALX receptor [99]2. Full-length protein or the N-terminal 26-amino acid peptide is anti-inflammatory [102]3. Serine protease inactivates anti-inflammatory properties [103]PGE21. EP3 -> activate mast cell -> ↑ permeability2. ↑ expression of IL23 -> ↑ Th17 differentiation [111]1. EP4 -> ↓ TNFα release [112]2. Change neutrophils into pro-resolution form -> ↓chemoattractant, ↑ lipoxin production [106]3. nuclear translocation of p50 ↑, p65 ↓ ; IkBa expression ↑ -> Inhibit NF-kB-mediated inflammatory signals [109]5-LOXPhosphorylation at ser271 and ser663 -> Perinuclear location -> leukotriene production [114]MerTK and resolvin D1 inhibit ser271 phosphorylation. Protein kinase A phosphorylate ser523 -> Cytoplasmic location -> SPM production [117]

## 5. The Release of Intracellular Molecules during Cell Death Processes

In the last decades, various regulated cell death pathways have been described, differing in triggering stimuli, signaling pathways, and immunological outcomes [5] (Table 3). Cell death processes vary in the mode, extent, and kinetics of cell membrane damage and, consequently, result in the release of different patterns of DAMPs and RAMPs. The differences in the pattern of molecules released during individual cell death pathways may indicate how cell death modalities affect inflammation and resolution.

Cell death and the release of intracellular molecules can occur due to pore formation in the plasma membrane or membrane rupture. Activation of ion-selective channels, which causes osmotic shock and leads to rupture of the cell membrane, is observed in necroptosis [118]. In the case of ferroptosis, osmoprotectants have been shown to delay this process [119]. In contrast, during pyroptosis, the dynamic opening-closing of the non-selective ion channels protects the cell membrane from complete rupture, enabling fine-tuned regulation of DAMP secretion and cytokine release [120,121]. Many necrotic cell death processes, pyroptosis, necroptosis, and ferroptosis are characterized by endosomal sorting complexes required for transport-III-mediated membrane repair. This has the ability to stop the cell death process in certain phases [122]. Ninjurin I-mediated pore formation has been described in necrosis and pyroptosis, but not in necroptosis [123]. These steps affect not only which DAMPs or RAMPs can be resolved, but also the sequence of their release.

In addition to cell membrane disruption changes in the permeabilization of nuclear, mitochondrial, endoplasmic reticulum, and lysosomal membranes can also determine which DAMPs are released during certain cell death processes [124]. Nuclei may contain different types of nuclear DAMPs, such as IL-1α, IL-33, HMGB1, histones, and genomic DNA [124]. The permeability of the nuclear envelope is increased at early stages of the apoptotic process [125], and dissolution of the nuclear membrane has been observed during netosis, as well [126]. However, nuclei remain intact during pyroptosis, [127] and unimpaired nuclei were also detected in the extracellular environment during necroptosis [128].

Various mitochondrial-derived DAMPs, such as N-formyl peptides, mtDNA, and mitochondrial transcription factor A can be released when the mitochondrial outer and inner membranes become permeable [129]; apoptosis is usually associated with mitochondrial outer membrane permeabilization. Importantly, this phenomenon also could be detectable upon non-apoptotic signals, including pyroptosis. Mitochondrial inner membrane permeability is a typical feature of necrotic cells, but also appears in the case of apoptosis, as well [130].

Apoptosis, autophagy-induced cell death, and necroptosis all lead to ER stress, subsequently inducing the elevated expression of various DAMPs as PKR-like endoplasmic reticulum kinase or calreticulin [33,131,132].

Lysosomal membrane permeabilization could be induced in netosis-, necroptosis- [133], pyroptosis- [127], and lysosome-dependent cell death [5]. Lysosomal secretion plays a role in DAMP release, for example, HMGB1, ATP and Cold-inducible RNA-binding protein. In addition, elevated ROS level and cathepsin release enable modifications of DAMPs, including HMGB1, IL-33, and calreticulin [8].

ATP depletion is characteristic of certain modes of cell death, such as necrosis, parthanatos, mitochondrial permeability transition, and cuproptosis [134,135]. The level of cellular ATP also decreases significantly in necroptotic cells [136]. Apoptosis is an ATP-requiring process, but, interestingly, its ATP reserves do not become depleted [137].

In the regulation of the balance between inflammation and resolution, signaling pathways operating in parallel with cell death signals can also be deterministic, in addition to cell death process-dependent DAMP production. Cytokines, chemokines, and inflammatory mediators can be produced by the signaling pathways that function simultaneously with cell death signals, and DAMPs can also undergo post-translational modifications due to these processes by oxidation or lysosomal cathepsin-dependent proteolysis [124]. For example, during apoptosis, caspase activity results in the blockage of many signaling pathways, including the ones leading to the production of inflammatory mediators: DAMPs, alarmins, and cytokines. Accordingly, DNA, HMGB1, IL-33, and interferons are directly inactivated by caspase-mediated cleavage. In contrast, translation is active until the last moment of plasma membrane permeabilization during necroptosis [138].

Several cell death pathways can be activated by PRR receptors, such as pyroptosis, necroptosis, netosis, and extrinsic apoptosis. As a logical consequence of these cell death processes, the release of DAMPs can enhance pathogen clearance by activating inflammation. PRRs can also generate classical immune reactions, including inflammatory cytokine secretion, simultaneously with cell death signals. As a result, cytokine and DAMP production triggers inflammation in an additive/synergistic manner [75]. We can assume that PRR-inducible cell death pathways have evolved to diminish the niche of pathogens. Accordingly, many pathogen escape mechanisms have been published concerning apoptosis, necroptosis, pyroptosis, and netosis [139,140]. The assumption that PRR-inducible cell death processes primarily evolved for pathogen clearance, rather than resolution, has been confirmed by experimental results. All necroptotic, pyroptotic, and netotic cell death evoke strong inflammatory responses, as observed in several human inflammatory disorders and various in vivo mouse models [1,141,142,143].

Other cell death pathways, such as intrinsic apoptosis, ferroptosis, parthanatos, mitochondrial permeability transition, entosis, oxeiptosis, or cuproptosis, are activated by various functional cell abnormalities. These abnormalities include an excessive increase in intracellular ROS or ion concentration, mitochondrial dysfunction, metabolic stress, and DNA damage [144]. The restoration of the perturbations in the cellular functions does not necessarily activate immune reactions, but rather requires tissue regeneration where RAMPs productions is expected. Accordingly, hardly any DAMP or cytokine release has been detected following these cell death modalities until now [1,145].
ijms-24-00016-t003_Table 3Table 3Brief summary of cell death pathways. We refer to [5] for deeper understanding.
Main TriggersBackbone of Signaling PathwayImmunological OutcomeResolution PotentialInnate ImmunityAdaptive ImmunityIntrinsic apoptosisprogrammed death, or death induced by various stress signals, such as DNA damage, oxidative stress, hypoxia, drugs, radiations.mitochondrial membrane permeabilization (MOMP), cytocrome C release induces apoptosome formation (APAF-1, CASP-9) leading to effector caspase activation (CASP3-6-7)typically anti-inflammatory [146] tolerogenic/immunogenic [147]PS, ANXA1 exposure, SPM, PGE2 producitonExtrinsic apoptosisdeath receptor activation or dependence receptor deactivation, related mechanisms are also activated by intracellular pathogensplasma membrane receptors triggers CASP8, which activates effector caspases, mainly CASP3-6-7inflammatory/anti-inflammatory stimulus and context dependent [146]tolerogenic/immunogenic [148] 
Secondary necrosisapoptotic stimuliinsufficient efferocytosis leads to disruption of the apoptotic cellsinflammatory/anti-inflammatory [149] Immunogenic [150]PS exposure, ANXA1, PGE2Necroptosispathogens, PRR, DR activation, drugs.activation of RIPK3 phosphorylates MLKL, which creates plasma membrane poresinflammatory [151]immunogenic [147]PS exposure, SPM, PGE2, IL-33 productionPyroptosis pathogens, especially intracellular bacteria, and non-infectious stimuli, DAMPs, ROS, ionscaspases 1 activation by inflammasome leads to cleavage of gasdermin D, which forms plasma membrane poreshighly inflammatory, IL-1β, and IL-18 production [152]immunogenic [147]PS exposure, PGE2 productionNetosispathogens, immunocomplexes, activated platelets, DAMPs, oxLDL, H2O2, mtDNAprotein kinase C and NAPDH oxidase activate mitogen-activated protein kinases, leading to the release of neutrophil extracellular trapinflammatory [153]tolerogenic/immunogenic [147]ANXA1 exposureFerroptosisimbalanced oxido-reduction system, lipid peroxidationblockade of cystine-glutamate antiporter or reduced activity of glutathione peroxidase (GPX4) results in lipid peroxidationinflammatory [154] tolerogenic/immunogenic[147]PGE2 productionParthanatosreactive oxygen and nitrogenspecies, hypoxiaPARP1 hyperactivation results in ATP depletion, increased MOMP, and, consequently, apoptosis inducing factor (AIF)-dependent and macrophage migration inhibitory factor (MIF)-induced DNA damageinflammatory [155]immunogenic[147]PS exposureMitochondrial permeability transition Increase in Ca ^2+^, K ^+^, ROS concentrationcyclophilin D-mediated cell death leading to mitochondrial intra membrane opening accompanied by depolarization, Ca release, matrix swelling, rupture of the mitochondrial outer membrane, and release of intermembrane proteins including cytochrome apoptosis versus necrosis may depend on enough ATP level with the possible occurrence of intermediate forms of death [156] immunogenic[147]
Lysosome-dependent cell deathintracellular pathogens, inflammationlysosomal membrane permeability allows cathepsin activation inflammatory [157]immunogenic [147]
Entosisdetachment from the matrix, antimitoticagentsengulfment of viable cellsin actomyosin-dependent way, with the contribution of adhesion proteins, RHOA family proteins or due to deregulated microtubule dynamicsthere is no data available on whether pro- or anti-inflammatory, but caspase-independent cell death [158]tolerogenic/immunogenic [147]
Oxeiptosis Elevated toxic ROS level, viruses, radiationsROS accumulation and mutations in ROS sensors, regulated by KEAP1, PGAM5 and AIFM1anti-inflammatory [159]tolerogenic [147]
Cuproptosiscopper-dependent death [6]intracellular Cu binds to lipoylated components in the tricarboxylic acid cycle, which leads to protein aggregation and proteotoxic stressInflammatory [135]potentially immunogenic [160]


## 6. Cell Death in Resolution

### 6.1. Apoptosis-Related Resolution

Currently, the efferocytosis of apoptotic cells is considered the most important signal for resolution. Recognition of phosphatidylserine (PS), a phospholipid localized in the inner plasma membrane leaflet, plays a key role during the phagocytic process. In the ATP-dependent flip-flop transport mechanism, PS can remain in the inner plasma membrane; however, following caspase-mediated inactivation of this flippase system, PS appears on the outer plasma membrane [161]. Following its externalization, PS creates eat-me signals that promote its recognition directly by various receptors and soluble molecules. The binding of Gas6 and Protein S bridging molecules to PS not only encourages phagocytosis, but also induces MerTK receptor activation and anti-inflammatory responses in the efferocytotic process [162]. Accordingly, PS-presenting liposomes mimic the apoptotic cells in terms of their anti-inflammatory effects, promoting angiogenesis and wound healing [163].

Widespread interrelated positive feedback mechanisms between M2 macrophage subpopulations, efferocytosis, MerTK receptor, and SPM production have been observed. Macrophage differentiation to the M2 phenotype increases their efferocytotic capacity. MerTK expression is mostly limited to M2 macrophages, and M2 macrophages produce higher amount of SPMs [164,165]. Efferocytosis of apoptotic bodies promotes macrophage differentiation to the M2 phenotype, which elevates IL4 and IL13 production, increases SPM biosynthesis, and induces MerTK activity [164,166]. MerTK signaling downregulates the pro-inflammatory cytokine production of macrophages. The expression of MerTK is required for increased efferocytosis and promotes lipoxin and resolvin synthesis by increasing non-phosphorylated, cytoplasmic 5-LOX forms [116,167]. SPM production enforces macrophage differentiation to the M2 phenotype, and elevated SPM levels can increase MerTK expression and the intensity of efferocytosis [164,168]. Thus, the increase of any of the components within this positive feedback loop can all initiate the conversion of pro-inflammatory signals into pro-resolution.

According to the widely acknowledged concept, the apoptosis and subsequent efferocytosis of neutrophil granulocytes recruited at the site of inflammation are the most important elements of resolution [169,170]. However, billions of circulating inactivated neutrophils die by apoptosis each day without inducing resolution, nor do intense apoptotic processes in the thymus or lymphatics lead to resolution [171]. This suggests that apoptosis alone is not sufficient to initiate resolution.

### 6.2. Necrotic Cell Death-Induced Resolution

Although the role of apoptotic cells in resolution has been proven by numerous studies [172], we cannot rule out the fact that other forms of cell death can also regulate the resolution process. Necrotic forms of cell death have also been observed to induce immune tolerance, even with the active DAMP release. This is presumably due to the dominance of simultaneous RAMP-SPM production [173]. Cell death is an indisputable precursor for resolution; however, limited data are available on the contribution of regulated necrotic cell death forms in resolution, and it has not been rigorously tested in relevant in vivo experimental models [174].

When the quantity of cell death exceeds the maximum capacity of efferocytosis, apoptosis switches to a special inflammatory form called secondary necrosis (comprehensive review in [150]). In in vitro studies and processes involving intense cell death in vivo, this type of necrotic cell death can also trigger resolution. Since caspase activity and the apoptotic pathway in the dying cells are both intact during secondary necrosis, the signal transduction pathways, as well as DAMP, cytokine, and chemokine production, are still partially downregulated. Even if not with sufficient intensity, efferocytosis of apoptotic cells also works at the same time. This results in a unique tissue environment favoring chronic inflammation, where efferocytosis-induced resolution and necrosis-induced DAMP production work together [150]. It is worth noting that differences between secondary necrosis and classical necrosis have been gathered, including lower ATP levels, but higher PGE2 and ANXA1 levels, IL-33 inactivation, HMGB1 oxidation, and increased cholesterol efflux leading to M2 differentiation. As we have highlighted, these listed factors can convert secondary necrosis from pro-inflammatory to anti-inflammatory [150] (Figure 2).

Currently, necrotic cell death is indirectly linked to resolution through the DAMP production-induced recruitment of neutrophil granulocytes and their subsequent apoptosis. PS-induced MerTK-mediated efferocytosis of apoptotic cells, especially apoptotic neutrophil granulocytes, is considered the most deterministic step of apoptosis-induced resolution. However, expression of PS has also been detected on pyroptotic and necroptotic cells, and the following parthanatos [175]. Thus, PS is not a specific ligand for apoptosis [176]. Consequently, efferocytosis-induced, MerTK-related SPM production and resolution potential can also be activated during regulated necrosis.

SPM production is typical for efferocytes engulfing apoptotic PMNs [47]. However, direct SPM release from dying cells has also been observed occasionally during apoptosis [1] and necroptosis [177]. Ferroptosis is mediated by lipid peroxidation, but the ERK and p38MAPK signaling pathways active during this process presumably block direct SPM production. To clarify this, further investigations would be necessary. Direct PGE2 production from dying cells upon various apoptotic [178], necroptotic [179], pyroptotic [179], and ferroptotic stimuli [180] has also been demonstrated. The release of PGE2 from necrotic cancer cells could be detected during chemotherapy, which can enhance tumor immune evasion [180]. Blocking PGE2 made gemcitabine-induced cancer cell death immunogenic [181], which increased the effectiveness of immunotherapy [182].

As highlighted previously, most DAMPs and RAMPs do not contain signal peptides; thus, passive release during cell death is one of the main ways for their secretion [8]. The release of DAMPs is inhibited during the removal of membrane-enveloped apoptotic bodies by phagocytosis. We can presume that the secretion of pro-resolution molecules (cytoplasmic, ER–derived, mitochondrial, or nuclear mediators) is inhibited simultaneously by their efficient packaging into apoptotic bodies and subsequent efferocytosis. Consequently, the liberation of SPMs and RAMPs is more likely to be seen during necrotic, rather than classical, apoptotic cell death.

ANXA1 was identified as part of the anti-inflammatory pattern of apoptotic neutrophils, triggering anti-inflammatory responses [183]. Surprisingly, ANXA1 was hardly exposed on the cell surface of primary apoptotic cells compared to secondary necrotic cells, the latter process effectively inhibiting proinflammatory cytokine production in macrophages [43,184]. ANXA1 expression also increased upon netosis [185], which can lead to resolution in the presence of macrophages or neutrophil granulocytes expressing FPR2. Release of other RAMPs has also been observed from necrotic cells during brain damage, with both αB-crystallin [186] and BiP [187] playing a role in resolution in the CNS.

Not only inflammatory mediators are inhibited by caspase-mediated cleavage during the apoptotic process, but also receptors with resolution potential. Downregulation of don’t-eat-me receptors, such as CD24, CD31, and CD47, during apoptosis has been published [188,189]. Importantly, negative signaling generated on phagocytic cells by these receptors may cooperate with the secreted DAMPs. As we mentioned, CD24 can bind DAMPs, such as HMGB1, HSP70, and HSP90, to Siglec G/10, converting their proinflammatory functions into anti-inflammatory activities. Together, these results suggest that resolution may be more intense during regulated necrosis than during apoptosis, unless the compensatory effect of the released DAMPs suppresses this response.

IL-33 expression is characteristic of necrotic cell death in certain cell types [190]. Necroptosis has been identified as a type of cell death associated with significant IL-33 release. Accordingly, necroptosis is often referred to as the dominant form of cell death forcing the differentiation of ILC2 cells and the polarization of M2 macrophages [191].

The early secretion of ATP was identified as a “find me” signal that attracted phagocytes and induced the silent clearance of dying cells; however, the simultaneous secretion of ATP and other DAMPs triggered inflammatory responses [192]. The ratio of extracellular ATP/AMP accumulation induced by necrotic cell death was correlated with the immunostimulatory capacity of dying cells [69]. Cell type-specific expression of ectonucleotidases and the amount of released ATP and adenosine are dependent on both the modality of cell death and tissue-specific factors, determining the anti- or pro-inflammatory reactions. Accordingly, the detection of ATP secretion does not necessarily indicate the inflammatory outcome of cell death; fine-tuning mechanisms can lead to significant differences in the ATP-dependent inflammatory reactions of necrotic cell death forms.

## 7. Conclusions

The intensity of cell death, the effectiveness of efferocytosis, the production of DAMPs and RAMPs, and the tissue environment all influence the balance of inflammation/resolution. It seems that necrotic cell death processes may play a major role in controlling RAMP and SPM release. However, due to the lack of studies, their exact role cannot currently be determined. With our article, we aimed to highlight the different mechanisms of each type of regulated cell death, and how they can induce different DAMP and RAMP releases. Thus, examining DAMP release alone cannot characterize the effect of cell death on the immune response. The effect of DAMP and RAMP balance can lead to inflammation or resolution and immunogenicity or tolerogenecity, but only complex investigation of these processes can lead to a thorough understanding of sterile inflammatory processes.

## Figures and Tables

**Figure 1 ijms-24-00016-f001:**
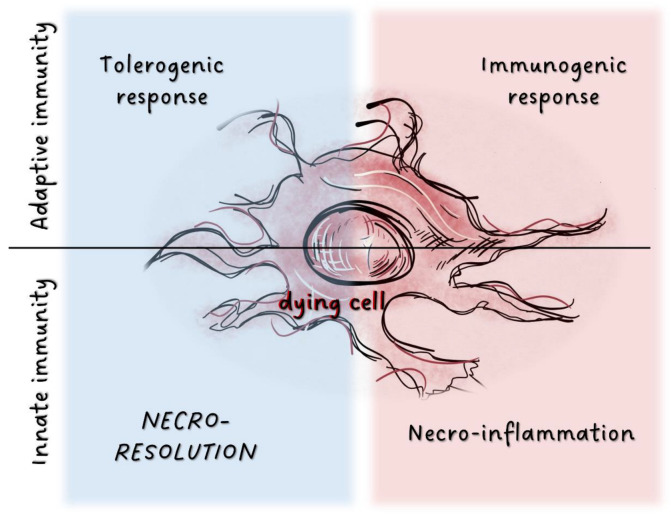
Potential immunological outcome of cell death processes.In adaptive immunity, factors released from dying cells are detected by various cell populations localized in tissues. This triggers either tolerogenic or immunogenic responses. Tissue damage results in the activation of the innate immune system through release of inflammatory mediators. Factors associated with dying cells can also induce immunosuppression as the inverse of inflammation, and dead cells may activate tissue resolution.

**Figure 2 ijms-24-00016-f002:**
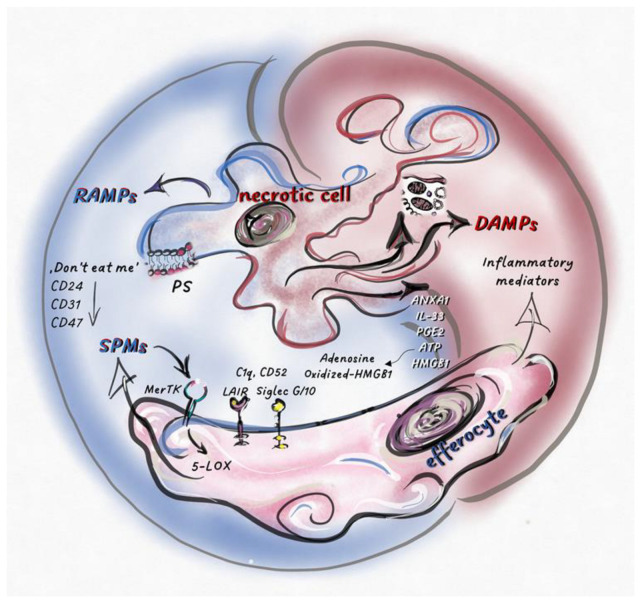
Necrotic cell death regulates the balance between pro-inflammatory- and pro-resolution-signals inducing necro-inflammation or necro-resolution.Intracellular molecules and molecular patterns associated with damage and resolution (DAMPs and RAMPs) are released from necrotic cells. Various cell types and their inflammatory or anti-inflammatory subpopulations sense these secreted mediators, together with dead cell corps in the tissue environment. Here, pro-inflammatory or pro-resolution mediators, such as SPMs, become dominant, which is determined by the type of cell death and the cells that sense it. Phosphatidylserine (PS) exposed on the surface of dying cells is recognized by MerTK receptors, stimulating the phosphorylation of 5-LOX in efferocytes and, consequently, increasing SPM production. “Don’t eat me” receptors (CD24, CD31, CD47) expressed by dying cells stimulate inhibitory receptors on efferocytes (such as Siglec 10), shifting the balance toward the anti-inflammatory response. The effects of the released factors could be modified by post-translational modifications or by the molecular partners interacting with them. After oxidation, HMGB1 loses its inflammatory potential, while its association with C1q, CD52, or CD24 results in its binding to inhibitory receptors (Siglec 10 or Leukocyte-associated immunoglobulin-like receptor 1 (LAIR1)), converting the effect of these complexes into anti-inflammatory. Extracellular ATP, which promotes inflammation, is degraded into immunosuppressive adenosine by cell surface receptors (CD39 and CD73). The pro-inflammatory or pro-resolution effect of IL-33, ANXA1, and PGE2 depends on the type of cells and/or receptors that perceive them.

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
