# Peer review of "Resolution Potential of Necrotic Cell Death Pathways"

_ijms, 2022, doi:10.3390/ijms24010016_

Round 1
Reviewer 1 Report
With this review, the authors aim to draw attention to the potential roles necrotic cell death could have in resolving inflammatory processes. As the authors rightly point out, a lot is known about how apoptotic cell death induces signaling processes that lead to the dampening of inflammation and the resolution of inflammation. In contrast, little is known about how the other cell death programs, that are by design more pro-inflammatory, contribute to resolution.
In this manuscript, the authors outline some examples of known intracellular proteins with pro-resolving functions, such as HSP10 and HSP27. They also nicely summarize the known functions of pro-resolving mediators, such as resolvins, in promoting the resolution of inflammatory processes in vivo. Furthermore, they also touch on the pro-and anti-inflammatory functions of known DAMPs with resolving potential, such at ATP and IL33. Here the authors try to point out the importance of timing, expression levels, and the cell-types affected by these signals for promoting resolution.
The authors also examine the potential mechanisms for resolution induced by specific cell death pathways (Table 3). Intriguingly, the mechanisms listed for pro-inflammatory pathways (such as pyroptosis, necroptosis and ferroptosis) would appear identical i.e., PS and Anxa1 exposure. Therefore, as the authors point out, the extent of tissue damage and the speed of resolution is most likely heavily dependent on the balance between pro-inflammatory and resolution mediators released by the dying cells, and on the tissue context. This aspect of resolution of inflammation downstream of these inflammatory cell death pathways is therefore an interesting direction for future research.
One negative comment about the manuscript is the un-necessary use of abbreviations that make the text sometimes hard to read, especially for the non-expert. I am not sure that the use of terms RAMPs and SMPs is really needed.
Author Response
Thank you very much for the reviewer's positive comments!
We agree with the reviewer that unnecessary abbreviations sometimes make the text difficult to read. We deleted 20 abbreviations that were used less than three times in the text. (PD1, RvE, LX, MaR, LTA4, PGD2, MPT, EAE, CL, SAA, mPGES-1, PG, CaMKII, PKA, ESCRT-III, PERK, CRT, SR A, ER, NET) We would still prefer to use the abbreviations SPM and RAMP, as they occur very often (>25) in the text.
Reviewer 2 Report
In their review, the authors comprehensively describe the resolution potential of necrotic cell deaths. This is an excellent review, easy-to-read and timely.
I would only advise the authors to have one last read to check for typos, as several can be found in their manuscript and to include considerations on how cuproptosis might also impact resolution.
Author Response
Thank you very much for the reviewer's positive comments and for bringing cuprotosis to our attention!
We included cuprotosis as an inflammatory cell death into Table 3 (10.1126/science.abf0529, 10.3389/fcell.2022.996307, 10.3389/fonc.2022.922332). In the list of necrotic and simultaneously controlled processes, cuproptosis is mentioned in line 49. Cuproptosis is listed as ATP-depleted cell death in line 441 10.3389/fcell.2022.996307.
The typos have been corrected.
Table 3. (completion)
|
Cuproptosis |
copper-dependent death 10.1126/science.abf0529 |
Intracellular Cu binds to lipoylated components in the tricarboxylic acid cycle, which leads to protein aggregation and proteotoxic stress. |
inflammatory 10.3389/fcell.2022.996307 |
potentially immunogenic 10.3389/fonc.2022.922332 |
Reviewer 3 Report
This review describes the role of DAMPS and RAMPs on inflammation and resolution in relation to different kinds of cell death.
Critic points:
1. Abstract: Space between words is missing in some parts. Furthermore, in the last sentence a word is missing. The sentence does not make sense like this.
2. Fig.1 is not really informative. Instead, authors should create a figure that presents the described signaling pathways in an overview. This would make it much easier for the readers to grasp the complex topic. Ideas for such a figure can be found in the cited literature from Klegeris or Shields et al.
3. Tab. 3: In the column “Innate Immunity” authors use the term “Necrotic”. However, “Necrotic” isn´t any term used for the innate immune system.
Author Response
Thank you very much for the reviewer's positive comments!
The typos have been corrected and the abstract has been improved.
We especially thank you for this constructive suggestions. We completed the manuscript with a new figure (Figure 2) showing how necrotic cell death regulates the balance of pro-inflammatory and pro-resolution signals.
We agrre with the reviewer, thus in table 3, regarding innate immunity, the word necrotic was changed to inflammatory in accordance with literature data.